# Unveiling the full picture of malnutrition in Sub-Saharan Africa: The extended composite index of anthropometric failure among children under-5 in the SDG era

**Amare Muche**[1]*, **Aznamariam Ayres**[1], **Tilahun Dessie Alene**[2],
**Robel Asaminew Mekonnen**[2], **Elsabeth Addisu**[3], **Gebeyehu Tsega**[4], **Yawkal Tsega**[5]

**1** Department of Epidemiology and Biostatistics, School of Public Health, College of Medicine and Health Sciences, Wollo University, Dessie, Ethiopia, **2** Department of Paediatrics, School of Medicine, College of Medicine and Health Sciences, Wollo University, Dessie, Ethiopia, **3** Department of Reproductive and Family Health, School of Public Health, College of Medicine and Health Sciences, Wollo University, Dessie, Ethiopia, **4** Department of Health Systems Management and Health Economics, School of Public Health, College of Medicine and Health Sciences, Bahir Dar University, Bahir Dar, Ethiopia, **5** Department of Health System and Management, School of Public Health, College of Medicine and Health Sciences, Wollo University, Dessie, Ethiopia

* amaremu7@gmail.com

## Abstract

### Introduction

The burden of malnutrition among children under-5 in Sub-Saharan Africa (SSA) remains a significant public health concern. Traditional indices such as stunting, wasting, and underweight often failing to capture the overlapping and multifaceted nature of malnutrition. The Extended Composite Index of Anthropometric Failure (ECIAF) offers a comprehensive measure by integrating stunting, wasting, underweight and obese/overweight addressing the overlap and co-occurrence of different forms of malnutrition conditions, thus providing a more accurate depiction of nutritional deficiencies. Aligning with the Sustainable Development Goals, the ECIAF serves as a robust tool for health policymakers and public health practitioners to identify high-risk populations, allocate resources effectively, and improve health outcomes for young children in this region. This study aimed to determine the pooled prevalence of the Extended Composite Index of Anthropometric Failure and to identify its associated factors among children under-5 in SSA countries.

### Methods

This study is a secondary data analysis of nationally representative community-based cross-sectional Demographic and Health Surveys (DHS) conducted in Sub-Saharan African countries, including a weighted sample of 176,141 children under-5 years. The recent demographic and health survey of 26 Sub-Saharan African countries were involved in this analysis. A multilevel binary logistic regression model was fitted to

**Data availability statement:** All relevant data necessary to replicate the study's findings are available within the manuscript and its Supporting Information files. The dataset used in this analysis is derived from the Demographic and Health Surveys (DHS) Program, which is a publicly accessible, third-party source. Due to data ownership and usage restrictions, we are not permitted to share the raw DHS dataset directly. However, the minimal data set underlying the results of this study—including the variables used, coding details, and values behind reported means, percentages, and regression analyses—are provided in the Supporting Information files. Researchers can obtain the original DHS data by registering for free and submitting a request through the DHS Program website: https://dhsprogram.com/data/. Access is granted to qualified researchers upon approval by the DHS Program. Data requests may be directed to: The DHS Program ICF, 530 Gaither Road, Suite 500, Rockville, MD 20850, USA Email: dhsprogram@icf.com.

**Funding:** The author(s) received no specific funding for this work.

**Competing interests:** The authors have declared that no competing interests exist.

**Abbreviations:** ANC: Antenatal Care; AOR: Adjusted Odds Ratio; CIAF: Composite Index of Anthropometric Failure; DIC: Deviance Information Criterion; DHS: Demographic and Health Survey; ECIAF: Extended Composite Index of Anthropometric Failure; EDHS: Ethiopian Demographic and Health Survey; ICC: Intra Class Correlation; LMICs: Low- and Middle-Income Countries; MOR: Median Odds Ratio; PCV: Proportional Change in Variance; SSA: Sub-Saharan Africa; SDG: Sustainable Development Goal; SSA: Sub-Saharan Africa; UNICEF's: United Nations Children's Fund; WHO: World Health Organization.

identify factors associated with ECIAF in children under-5 years old. The AOR with its 95% CI was estimated, and a level of significance of 0.05 was used to determine the significant factors of ECIAF.

## Results

The pooled prevalence of ECIAF among children under-5 was 36% (95% CI: 33%, 40%) in Sub-Saharan Africa. Key factors associated with increased odds of ECIAF included: increasing child age (6–23 months [AOR = 1.56; 95% CI: 1.46, 1.66] & 24–59 months [AOR = 2.03; 95% CI: 1.88–2.19]), multiple births [AOR = 2.38; 95% CI: 2.05–2.76], reducing birth size (average [AOR = 1.20; 95% CI: 1.14, 1.26] & small [AOR = 1.80; 95% CI: 1.68–1.93]), having comorbidity [AOR = 1.12; 95% CI: 1.07, 1.16], reducing level of mother's educational status (primary [AOR = 1.20; 95% CI: 1.12, 1.28] & no formal education [AOR = 1.36; 95% CI: 1.27, 1.45]), increasing number of children under-5 in the household (2 children [AOR = 1.17; 95% CI: 1.10, 1.23] & 3–5 children [AOR = 1.14; 95% CI: 1.06, 1.22]), reducing household wealth status (rich [AOR = 1.28; 95% CI: 1.17, 1.41], middle [AOR = 1.31; 95% CI: 1.19, 1.44], poor [AOR = 1.42; 95% CI: 1.28, 1.56] & poorest [AOR = 1.45; 95% CI: 1.31, 1.61]), living in rural area [AOR = 1.15; 95% CI: 1.09, 1.22]. Protective factors included female sex [AOR = 0.69; 95% CI: 0.66–0.72], birth interval >24 months [AOR = 0.86; 95% CI: 0.81, 0.91] health facility delivery [AOR = 0.79; 95% CI: 0.74–0.83], and antenatal care attendance [AOR = 0.83; 95% CI: 0.77–0.90].

## Conclusions

The pooled prevalence of ECIAF in children under-5 was high in Sub-Saharan Africa. Addressing this burden requires scaling up nutrition-sensitive interventions that tackle underlying determinants of child malnutrition, including poverty reduction, women's education and empowerment, improved access to quality health services, safe water, sanitation, and hygiene programs. Such multisectoral strategies, alongside maternal and child health interventions, are essential to mitigate the identified risk factors and sustainably reduce ECIAF in the region.

## Introduction

The prevalence of malnutrition among children under-5 remains a significant public health concern in Sub-Saharan Africa (SSA). This manuscript introduces an innovative approach to assessing child malnutrition using the Extended Composite Index of Anthropometric Failure (ECIAF). Traditional indices such as weight-for-age, height-for-age, and weight-for-height have been instrumental in monitoring child nutrition, but they often fail to capture the multifaceted nature of malnutrition. The ECIAF offers a more comprehensive measure by integrating multiple dimensions of anthropometric failure, thereby providing a more complete assessment of the nutritional status of young children in this region [1–4].

Malnutrition in children under-5 manifests in various forms, including stunting (low height-for-age), wasting (low weight-for-height), and underweight (low weight-for-age). These conditions are not only markers of immediate nutritional deficiencies but also predictors of long-term health outcomes, cognitive development, and economic productivity [5–7]. According to the World Health Organization (WHO), over one-third of child deaths are attributable to undernutrition, highlighting the urgency of addressing this issue. The ECIAF aims to capture the overlap and co-occurrence of these anthropometric failures, providing a more accurate depiction of the burden of malnutrition [1,2].

The Sustainable Development Goals (SDGs), particularly Goal 2 (Zero Hunger) and Goal 3 (Good Health and Well-being), prioritize the eradication of all forms of malnutrition and the promotion of healthy lives for all. The ECIAF aligns directly with these goals by offering a comprehensive tool to identify the true burden of child malnutrition, offering a robust tool for policymakers and health practitioners, thereby enabling more accurate targeting of interventions and resource allocation to populations most in need, ultimately supporting progress toward achieving these global health targets [8].

Previous research has extensively documented the prevalence and determinants of malnutrition using traditional indices. However, these conventional indicators often assess malnutrition in isolation, overlooking the overlapping and coexisting forms of nutritional deficits in the same child; this gap necessitates a more integrated approach such as the Extended Composite Index of Anthropometric Failure (ECIAF). For instance, a child can be both stunted and wasted, a condition that might be missed if only one index is used [9,10]. The ECIAF addresses this research gap by considering the composite nature of anthropometric failure, thereby revealing the true extent of nutritional deficiencies in children under-5 [4].

The development of the ECIAF is grounded in a comprehensive review of existing literature and empirical data from national health surveys in Sub-Saharan Africa. This index incorporates height-for-age, weight-for-age, and weight-for-height measurements into a single composite score, facilitating a more holistic assessment of child nutrition. By capturing multiple dimensions of malnutrition, the ECIAF provides a clearer picture of the nutritional landscape and helps identify children who are at the greatest risk of adverse health outcomes.

One of the critical gaps identified in previous studies is the absence of a standardized composite measure that can be applied across different contexts and populations. While traditional indices are useful, they often fail to capture the complex interplay between various forms of malnutrition. The ECIAF fills this void by offering a standardized tool that can be used across different regions and demographic groups, thereby enhancing the comparability of data and facilitating more effective interventions.

The health policy relevance of the ECIAF cannot be overstated. By providing a more accurate and comprehensive measure of child malnutrition, this index can inform the design and implementation of targeted nutrition programs. Policymakers can use ECIAF data to identify regions with the highest burden of malnutrition and allocate resources accordingly. Furthermore, the ECIAF can help monitor the impact of nutrition interventions over time, providing valuable feedback for program improvement.

From a practical standpoint, the ECIAF offers several advantages over traditional indices. Health practitioners can use this composite measure to conduct more thorough nutritional assessments and identify children who require immediate intervention. The ECIAF can also serve as a useful tool for community health workers and non-governmental organizations involved in nutrition and child health programs. By providing a more detailed understanding of the nutritional status of children under-5, the ECIAF can help improve the targeting and effectiveness of nutrition interventions.

In conclusion, the Extended Composite Index of Anthropometric Failure represents a significant advancement in the assessment of child malnutrition in Sub-Saharan Africa. By integrating multiple dimensions of anthropometric failure, the ECIAF provides a more comprehensive and accurate measure of nutritional deficiencies [4]. This index aligns with the objectives of the Sustainable Development Goals and addresses critical gaps in existing research. The ECIAF holds substantial promise for informing health policy, guiding nutrition interventions, and ultimately improving the health and well-being of children under-5 in this region.

However, the utility of ECIAF also depends on the availability of high-quality anthropometric data and the technical capacity to compute and interpret its components accurately. The index's complexity may limit its application in routine programmatic settings without adequate training or digital tools. Furthermore, although ECIAF has shown promise in research settings, its broader validation across diverse populations and its predictive value for long-term child health outcomes warrant further investigation. Therefore, this study aimed to determine the pooled prevalence of the extended composite index of anthropometric failure and to identify its associated factors among children under-5 in SSA countries.

## Methods

### Study settings, population and data sources

This research utilized the most recent Demographic and Health Survey (DHS) data from 26 Sub-Saharan African (SSA) countries that had DHS surveys conducted between 2015 and 2022. The countries included in this study are Kenya, Ethiopia, Madagascar, Rwanda, Tanzania, Uganda, Mozambique, Benin, Burkina Faso, Côte d'Ivoire, Gambia, Ghana, Guinea, Liberia, Mali, Nigeria, Senegal, Sierra Leone, Burundi, Cameroon, Gabon, Mauritania, Angola, Malawi, South Africa, and Zambia (S1 Table).

Demographic and Health Survey is a nationally representative survey conducted every five years that gathers data on key health indicators, including maternal and child health, fertility, health care system, morbidity, and mortality characteristics. The survey employs a two-stage stratified cluster sampling method: in the first stage, Enumeration Areas (EAs) were selected from each country using probability proportional to size sampling (PPS) based on the country's latest population figures and housing census. The stratification was based primarily on geographic regions (e.g., provinces, states, or zones, depending on the country) and residence type (urban vs. rural), resulting in distinct strata for sampling within each country. In the second stage, a fixed number of households were randomly selected within each selected EA. This stratified approach allowed for adequate representation of various geographic and socio-demographic contexts, ensuring that estimates could be generalized across different subpopulations within countries.

Each country's survey includes various datasets for children, women, men, births, and households. This study is a secondary data analysis of the most recent Demographic and Health Surveys (DHS) conducted between 2015 and 2022 across 26 Sub-Saharan African countries. For this analysis, we used the Kids Record (KR) datasets, focusing on 176,141 children under-5 years of age with valid anthropometric measurements. Further information on DHS methodology can be found at the official database: https://dhsprogram.com/Methodology/index.cfm and from the respective 26 countries of the most recent DHS reports.

### Measurement variables

**Dependent variable.** The outcome variable in this study was children's "extended composite index of anthropometric failure" assessed using W/H<−2 Z-score, W/H>+2 Z-score, H/A<−2 Z-score and W/A<−2 Z-score with respect to the standards of the World Health organization (WHO). Extended composite index of anthropometric failure **=1 if a child falls into any failure group (B to H), otherwise 0 (group A).**

**Level stratifier variable.** The stratifier variable was the cluster number or enumeration area/EA (V001).

**Independent variables.** The explanatory variables were divided into community-level and individual-level determinants. Community-level determinants and individual-level determinants such as African region of the country, child place of residence, child characteristics, maternal characteristics, socioeconomic, sociodemographic, household characteristics were selected based on previous studies in similar settings and their availability in the DHS datasets. The community-level determinants included for this study were the Sub-Saharan African region (East, West, Central, South and North), the child place of residence (urban or rural) community-level women's education and community-level poverty. The individual-level determinants included for the current study were mother's education, age and marital

status, child age, sex, and size at birth, birth type, birth order, preceding birth interval, parity, place of delivery, antenatal care visits, family size, number of under-five children, wealth index, time of breastfeeding initiation, media exposure, presence morbidity, and vaccine status. The variables were categorized based on the previous literature and scientific backgrounds.

The community-level determinants; Community-level women's education, and community-level poverty were created by aggregating individual-level observations at the cluster level, using average proportions of women in each category of a variable and categorizing them into low and high groups based on median values.

Media exposure was defined by three variables: frequency of listening to the radio, watching television, and reading newspapers or magazines. Women who engaged in any of these activities at least once a week were considered to have media exposure (coded 'yes'), while others were coded as not having media exposure ('no').

**Operational definitions. Extended composite index of anthropometric failure (ECIAF)** is an anthropometric index that combines weight-for-age (WAZ), length/height-for-age (HAZ) and weight-for-length (WHZ) to determine the nutritional status of children under-5 years old. The detailed categories of ECIAF are grouped in to nine which includes: (A) Without anthropometric failure; (B) Wasting only; (C) Wasting and underweight; (D) Wasting, stunting and underweight; (E) stunting and underweight; (F) stunting only; (Y) underweight only; (G) weight excess (being overweight or obese) only and (H) stunting and weight excess (Table 1). The anthropometric failure is the total amount of malnutrition or sum of category of wasting only; wasting and underweight; wasting, stunting and underweight; stunting and underweight; stunting only; underweight only; weight excess (being overweight or obese) only; and stunting and weight excess. At the same time, the ECIAF index can be used to detect some anthropometric failures [4].

**Africa region** For the current study the African continent is divided into five regional groupings. **Northern African region** includes Mauritania. The **Southern African region** comprises Angola, Malawi, South Africa, and Zambia. The **Western African region** encompasses Benin, Burkina Faso, Côte d'Ivoire, Gambia, Ghana, Guinea, Liberia, Mali, Nigeria, Senegal, and Sierra Leone. The **Eastern African region** includes Ethiopia, Kenya, Madagascar, Rwanda, Tanzania, Uganda, and Mozambique. Finally, the **Central African region** comprises Burundi, Cameroon, and Gabon.

**Community Poverty level** The household wealth status of the community below the median was considered as poor or greater/equal to the median was considered as a rich community wealth index.

**Community women literacy** Community considered as literate if at least 50% of women in the community attend at least primary education and illiterate if women in the community had no education or only less than half proportion of women in the community educated.

**Table 1. Categories of the Extended Composite Index of Anthropometric (ECIAF).**

| Categories | Description | W/H or BMI/A<−2SD | W/H or BMI/A>2SD | H/A<−2SD | W/A<−2SD |
|---|---|---|---|---|---|
| A | Without anthropometric failure | No | No | No | No |
| B | Wasting only | Yes | No | No | No |
| C | Wasting and underweight | Yes | No | No | Yes |
| D | Wasting, stunting and underweight | Yes | No | Yes | Yes |
| E | Stunting and underweight | No | No | Yes | Yes |
| F | Stunting only | No | No | Yes | No |
| Y | Underweight only | No | No | No | Yes |
| G | Weight excess (being overweight or obese) only | No | Yes | No | No |
| H | Stunting and weight excess | No | Yes | Yes | No |

ECIAF = % of children falling into any of the categories B through H

**Source:** The proposal of Extended Composite Index of Anthropometric Failure (ECIAF) by Bejarano IF, Oyhenart EE, Torres MF, Cesani M, Garraza M, Navazo B, et al. in 2019 (4).

## Sampling procedures and data collection period

*The surveys utilized a two-stage stratified cluster sampling technique to ensure nationally representative estimates across 26 Sub-Saharan African (SSA) countries.* In the first stage, strata were defined by combining geographical regions within each country (e.g., administrative regions, provinces, or states) and place of residence (urban vs. rural). Each country was first divided into these strata, ensuring variation across both geographic and settlement types.

From each stratum, Enumeration Areas (EAs) were selected using probability proportional to size (PPS) sampling, based on the most recent population and housing census data. In the second stage, a predetermined number of households (usually 20–30 per EA) were systematically selected from updated household listings within each EA. This design allows for the inclusion of diverse population subgroups across socioeconomic and geographic contexts, ensuring the sample reflects both national and regional-level variation in child health and nutrition indicators.

This stratified approach, common to all DHS surveys, enhances the representativeness of the data and facilitates disaggregated analysis by region and residence type (urban/rural), which is crucial for equity-focused health research and planning.

The data were collected between 2015 and 2022, aligning with the Sustainable Development Goals (SDG) era, which began in 2015. The most recent Demographic and Health Survey (DHS) data available from each of the 26 Sub-Saharan African countries during this period were used in the analysis.

## Data management and analysis

In this study data from 26 Sub-Saharan African (SSA) countries were cleaned, appended, recoded, and analyzed using Stata V.17 MP Software. Data were weighted prior to analysis using sampling weights, primary sampling units, and stratification variables to ensure national representativeness. Descriptive statistics were presented in text, tables and graphs. To account for the clustering effects of demographic and health survey (DHS) data, a mixed effect analysis was applied to determine the effects of independent variable on the dependent variable (extended composite index of anthropometric failure (ECIAF)). Bivariate multilevel binary logistic regression analysis was carried to identify variables eligible for the multivariable analysis. Variables with a **p-value<0.2** in the bivariate analysis were considered candidates for the multivariable multilevel model. In addition, **prior empirical findings from peer-reviewed literature** and insights from **subject-matter experts in maternal and child health** guided the inclusion of key variables, even if they did not meet the bivariate threshold. By using these techniques, selected variables were entered to the multivariable multilevel binary logistic regression model.

Before the final model building, Intraclass correlation (ICC), median odds ratio (MOR), and proportional change of variance (PCV) were computed to measure the variation between clusters. The ICC measures the degree of heterogeneity between clusters (EAs) [11].

$$\text{ICC} = \sigma^2/(\sigma^2 + \pi^2/3)$$

$$\text{MOR} = \exp\sqrt{2 \times \partial^2 \times 0.6745} \sim \text{MOR} = \exp(0.95 \times \partial)$$

The likelihood ratio (LR) test, intraclass correlation coefficient (ICC), and median odds ratio (MOR) were calculated to measure the variation between clusters. The ICC quantifies the degree of heterogeneity between clusters, representing the proportion of total observed individual variation in malnutrition among children under-5. The MOR quantifies the variation in malnutrition between clusters on the odds ratio scale and is defined as the median value of the odds ratio between a high-risk cluster and a low-risk cluster when randomly selecting individuals from two clusters (Enumeration Areas, EAs).

**Proportional Change in Variance (PCV)** is a measure used in multilevel modeling to quantify the proportion of cluster-level variance explained by the addition of predictors in successive models. It is calculated by comparing the variance of the null model (with no predictors) to the variance in subsequent models.

**Mathematically:**

**PCV = (Variance in Null Model – Variance in Model X)/ Variance in Null Model**

A higher PCV indicates that more of the **between-cluster variability** in the outcome (ECIAF) is explained by the included covariates. In this study, the PCV helped assess how much of the variation in ECIAF was accounted for by individual-level and community-level factors in the full model.

After selecting the variables for multivariable multinomial analysis, four models were fitted to identify the best-fitting model: Model I (null model with only the outcome variable), Model II (only individual-level explanatory variables), Model III (only community-level explanatory variables), and Model IV (full model examining both individual and community-level predictors). The best-fitting model was selected using log-likelihood ratio (LLR) and deviance (−2LLR) values, with the model having the lowest deviance considered the best fit. Statistical significance of the independent variables was declared with a p-value < 0.05 and the adjusted odds ratio (AOR) with a 95% confidence interval (CI). Collinearity diagnostics showed that the variance inflation factor values for all variables in the final model were below 10, indicating no multicollinearity. Additionally, ICC and MOR were computed to estimate the random variability in ECIAF status between clusters.

### Ethical considerations

This study is based on a secondary analysis of publicly available Demographic and Health Survey (DHS) data. Permission to access and use the datasets was obtained through registration and approval from the DHS Program website (https://dhsprogram.com) (S2 File). The DHS data are de-identified and collected with informed consent from participants by the respective national authorities. As such, no additional ethical approval was required for this analysis.

## Results

### The study population characteristics

Weighted samples of 176,141 children under-5 years old were included in the analysis. A total of 26 African countries were considered and categorized in five regions which were including seven East African countries category such as Ethiopia, Kenya, Rwanda, Uganda, Tanzania, Mozambique and Madagascar; eleven Western African countries such as Benin, Burkina Faso, Côte d'Ivoire, Gambia, Ghana, Guinea, Liberia, Mali, Nigeria, Senegal, and Sierra Leone; three Central African countries such as Burundi, Cameroon, and Gabon; one Northern African country (Mauritania) and four Southern African countries such as Angola, Malawi, Zambia and South Africa.

From the study participants more than half (50.51%) of the children were males. Age range of children under-5 years was 0−59 months, and the mean age was 28.23 (SD = 17.3) months. After recoding child age, the majorities (57.09%) were 24 months and above, 31.79% were 6−23 months and 11.12% were below 6 months. Nearly two-thirds (64.09%) children under-5 years were from rural residents in Sub-Saharan Africa. About 41.53% of the children under-5 years were in Western Africa Region, 28.32% were in Eastern Africa Region, 13.31% were in Southern African region, 10.41% were in Central Africa Region, and the rest 6.42% were in Northern Africa Region in Mauritania (Table 2).

### Prevalence of ECIAF by study characteristics in children under-5 in Sub-Saharan Africa in the SDG era

The distribution of extended composite index of anthropometric failure varies in characteristics of child, maternal household and community variables in children under-5 years in Sub-Saharan African countries (Table 3).

### The pooled prevalence of ECIAF failure in children under-5 in Sub-Saharan Africa in the SDG era

The pooled prevalence of extended composite index of anthropometric failure (ECIAF) in children under-5 years old in Sub-Saharan Africa (SSA) was 36% with a 95% confidence interval ranging from 33% to 40%. Prevalence ranged from 59% in Burundi (95% CI: 58–60%) to 20% in Gabon (95% CI: 19–21%) (Fig 1).

**Table 2. Study characteristics.**

| Variables (categories) | Count | Percentage (%) |
|---|---|---|
| **Children characteristics** | | |
| **Child age (months)** | | |
| <6 months | 17,181 | 11.12 |
| 6-23 months | 49,108 | 31.79 |
| 24-59 months | 88,198 | 57.09 |
| **Child sex** | | |
| Male | 78,033 | 50.51 |
| Female | 76,454 | 49.49 |
| **Birth order** | | |
| 1st child | 34,495 | 22.33 |
| 2nd or 3rd child | 56,859 | 36.81 |
| 4th or above | 63,132 | 40.87 |
| **Type of birth** | | |
| Single birth | 149,642 | 96.86 |
| Multiple birth | 4,845 | 3.14 |
| **Preceding birth interval of child in months** | | |
| <24 months | 20,742 | 17.34 |
| >=24 months | 98,908 | 82.66 |
| **Breast feeding initiation** | | |
| Recommended | 84,588 | 62.99 |
| not recommended | 49,705 | 37.01 |
| **Perceived child size at birth** | | |
| Large | 38,2321 | 30.09 |
| Average | 67,694 | 53.28 |
| Small | 21,132 | 16.63 |
| **Child has Comorbidity** | | |
| No | 94,562 | 63.24 |
| Yes | 54,973 | 36.76 |
| **Child vaccination status** | | |
| Completely immunized | 38,810 | 40.47 |
| Incompletely immunized | 57,087 | 59.53 |
| **Maternal characteristics** | | |
| **Age of the mother in years** | | |
| 15-24 years | 41,630 | 26.95 |
| 25-34 years | 75,069 | 48.59 |
| 35-49 years | 37,788 | 24.46 |
| **Educational level of the mother** | | |
| Secondary education and above | 47,240 | 30.58 |
| Primary education | 50,026 | 32.38 |
| Has no formal education | 57,215 | 37.04 |
| **Current marital status of the mother** | | |
| Married | 140,229 | 90.77 |
| Single | 9,861 | 6.38 |
| Divorced | 2,649 | 1.71 |
| Widowed | 1,748 | 1.13 |

*(Continued)*

**Table 2.** (Continued)

| Variables (categories) | Count | Percentage (%) |
|---|---|---|
| **Parity** | | |
| Primi parity | 26,170 | 16.94 |
| Multi parity | 62,910 | 40.72 |
| Grand parity | 65,407 | 42.34 |
| **Place of delivery** | | |
| Home delivery | 35,185 | 23.37 |
| Health institutional delivery | 115,387 | 76.63 |
| **Mother's antenatal visit during pregnancy** | | |
| Has no antenatal visit | 8,490 | 8.29 |
| Has antenatal visit | 93,976 | 91.71 |
| **Household characteristics** | | |
| **Number of household members** | | |
| 1-4 family members | 39,017 | 25.26 |
| 5-9 family members | 87,417 | 56.59 |
| >=10 family members | 28,053 | 18.16 |
| **Number of children under-5 in the household** | | |
| 1 child | 51,440 | 33.30 |
| 2 children | 63,046 | 40.81 |
| >=3 children | 40,001 | 25.89 |
| **Wealth index of the household** | | |
| Richest | 26,370 | 17.07 |
| Richer | 29,585 | 19.15 |
| Middle | 30,886 | 19.99 |
| Poorer | 32,571 | 21.08 |
| Poorest | 35,075 | 22.70 |
| **Community level characteristics** | | |
| **Place of residence** | | |
| Urban | 55,481 | 35.91 |
| Rural | 99,006 | 64.09 |
| **Media exposure** | | |
| Yes | 101,793 | 68.07 |
| No | 47,742 | 31.93 |
| **African regions** | | |
| Northern Africa | 9,916 | 6.42 |
| Southern Africa | 20,558 | 13.31 |
| Western Africa | 64,165 | 41.53 |
| Eastern Africa | 43,758 | 28.32 |
| Central Africa | 16,090 | 10.41 |

### Subgroup analysis of prevalence of ECIAF in children under-5 by African region in the SDG era

Subgroup analyses revealed notable regional variations in ECIAF prevalence across Sub-Saharan Africa. In East Africa, the prevalence was 37% (95% CI: 29%–44%), with Madagascar reporting the highest rate at 46% (95% CI: 45%–48%) and Kenya the lowest at 23% (95% CI: 23%–24%). In West Africa, the overall prevalence was 33% (95% CI: 30%–37%), with Nigeria at the high end (42%; 95% CI: 42%–43%) and both Ghana and Gambia at the low end (24%; 95% CI: 23%–25% and 23%–26%, respectively). Southern Africa had a prevalence of 41% (95% CI: 40%–43%), with Angola at

Table 3. The prevalence of ECIAF by study characteristics in children under-5 years in SSA.

| Variable | Categories | ECIAF | |
|---|---|---|---|
| | | Yes | No |
| Child characteristics | | n (%) | n (%) |
| Sex of child | Male | 48,030 | 30,004 |
| | Female | 51,060 | 25,393 |
| Age of child in months | <6 months | 4,866 | 12,315 |
| | 6-23 months | 17,753 | 31,355 |
| | 24-59 months | 32,779 | 55,419 |
| Birth order | 1st child | 11,720 | 22,775 |
| | 2nd or 3rd child | 19,468 | 37,391 |
| | 4th or above | 24,209 | 38,924 |
| Type of birth | Single birth | 52,858 | 96,784 |
| | Multiple birth | 2,539 | 2,306 |
| Preceding birth interval of child in months | <24 months | 8,761 | 11,982 |
| | >=24 months | 34,743 | 64,166 |
| Breast feeding initiation | recommended | 31,183 | 53,405 |
| | not recommended | 18,338 | 31,366 |
| Perceived child size at birth | Large | 12,676 | 25,556 |
| | Average | 24,766 | 42,927 |
| | Small | 9,500 | 11,632 |
| Child has Comorbidity | No | 32,694 | 61,867 |
| | Yes | 20,552 | 34,421 |
| Child vaccination status | Completely immunized | 14,580 | 24,229 |
| | Incompletely immunized | 20,306 | 36,782 |
| Maternal characteristics | | | |
| Age of the mother in years | 15-24 years | 15,768 | 25,862 |
| | 25-34 years | 26,323 | 48,746 |
| | 35-49 years | 13,306 | 24,481 |
| Educational level of the mother | Secondary education and above | 12,476 | 34,764 |
| | Primary education | 19,109 | 30,917 |
| | Has no formal education | 23,809 | 33,406 |
| Current marital status of the mother | Married | 50,327 | 89,901 |
| | Single | 3,353 | 6,507 |
| | Divorced | 1,025 | 1,624 |
| | Widowed | 692 | 1,056 |
| Parity | Primi parity | 8,806 | 17,364 |
| | Multi parity | 21,694 | 41,216 |
| | Grand parity | 24,897 | 40,509 |
| Place of delivery | Home delivery | 16,129 | 19,056 |
| | Health institutional delivery | 37,784 | 77,604 |
| Mother's antenatal visit during pregnancy | Has no antenatal visit | 3,980 | 4,510 |
| | Has antenatal visit | 32,568 | 61,408 |
| Number of household members | 1-4 family members | 13,491 | 25,525 |
| | 5-9 family members | 31,804 | 55,613 |
| | >=10 family members | 10,101 | 17,951 |

*(Continued)*

**Table 3.** (Continued)

| Variable | Categories | ECIAF | |
|---|---|---|---|
| | | **Yes** | **No** |
| **Number of children under-5 in the household** | 1 child | 16,662 | 34,777 |
| | 2 children | 23,278 | 39,767 |
| | >=3 children | 15,456 | 24,544 |
| **Wealth index of the household** | Richest | 6,006 | 20,363 |
| | Richer | 9,188 | 20,396 |
| | Middle | 11,198 | 19,687 |
| | Poorer | 13,284 | 19,287 |
| | Poorest | 15,719 | 19,354 |
| **Community level characteristics** | | | |
| **Place of residence** | Urban | 15,510 | 39,969 |
| | Rural | 39,886 | 59,120 |
| **Media exposure** | Yes | 32,341 | 69,451 |
| | No | 20,905 | 26,836 |
| **African regions** | Northern Africa | 3,229 | 6,688 |
| | Southern Africa | 8,552 | 12,006 |
| | Western Africa | 22,186 | 41,979 |
| | Eastern Africa | 14,866 | 28,892 |
| | Central Africa | 6,565 | 9,524 |

44% (95% CI: 42%–45%) and South Africa at 38% (95% CI: 36%–45%). In Central Africa, the prevalence was 39% (95% CI: 16%–63%), with Burundi exhibiting the highest rate among all countries at 59% (95% CI: 58%–60%), while Gabon recorded the lowest at 20% (95% CI: 19%–21%). These findings highlight substantial regional disparities in child malnutrition and underscore the need for geographically tailored nutrition interventions (Fig 2).

### Factors associated with ECIAF in children under-5 in Sub-Saharan Africa

**Fixed effect analysis.** Bi-variable analysis was performed to identify potential candidate variables for the multivariable analysis using 0.2 level of significance. In addition, literature review and expert consultation was considered for the final multivariable analysis. A null model (model I), a model with individual level variables (model II), a model with community level variables (model III) and a full model both with individual and community level variables (model IV) were performed to identify significantly associated variables with extended composite index of anthropometric failure (ECIAF) in children under-5 years in SSA.

From the final multilevel model, the risk of having ECIAF was higher among children who are 6−23 months and 24−59 months compared with those who are less than 6 months (AOR: 1.56; 95% CI: 1.46, 1.66, AOR: 2.03; 95% CI: 1.88, 2.19) respectively keeping the other covariates constant. Being female was less likely to have ECIAF compared with those who are being male (AOR: 0.69; 95% CI: 0.66, 0.72). The odds of having ECIAF was higher among children under-5 with multiple birth type (AOR: 2.38; 95% CI: 2.05, 2.76) as compared with those single birth type. The odds of having ECIAF was lower among children having long birth spacing (greater than 24 months) (AOR: 0.86; 95% CI: 0.81, 0.91) as compared to children having short birth spacing (less than 24 months) keeping the other covariates constant. The odds of having ECIAF was higher among children under-5 having perceived average birth size (AOR: 1.20; 95% CI: 1.14, 1.26) and perceived small birth size (AOR: 1.80; 95% CI: 1.68, 1.93) as compared with children whose birth size was perceived as large. The odds of having ECIAF was higher among children under-5 who had comorbidity (AOR: 1.12; 95% CI: 1.07, 1.16) as compared with those who had no comorbidity.

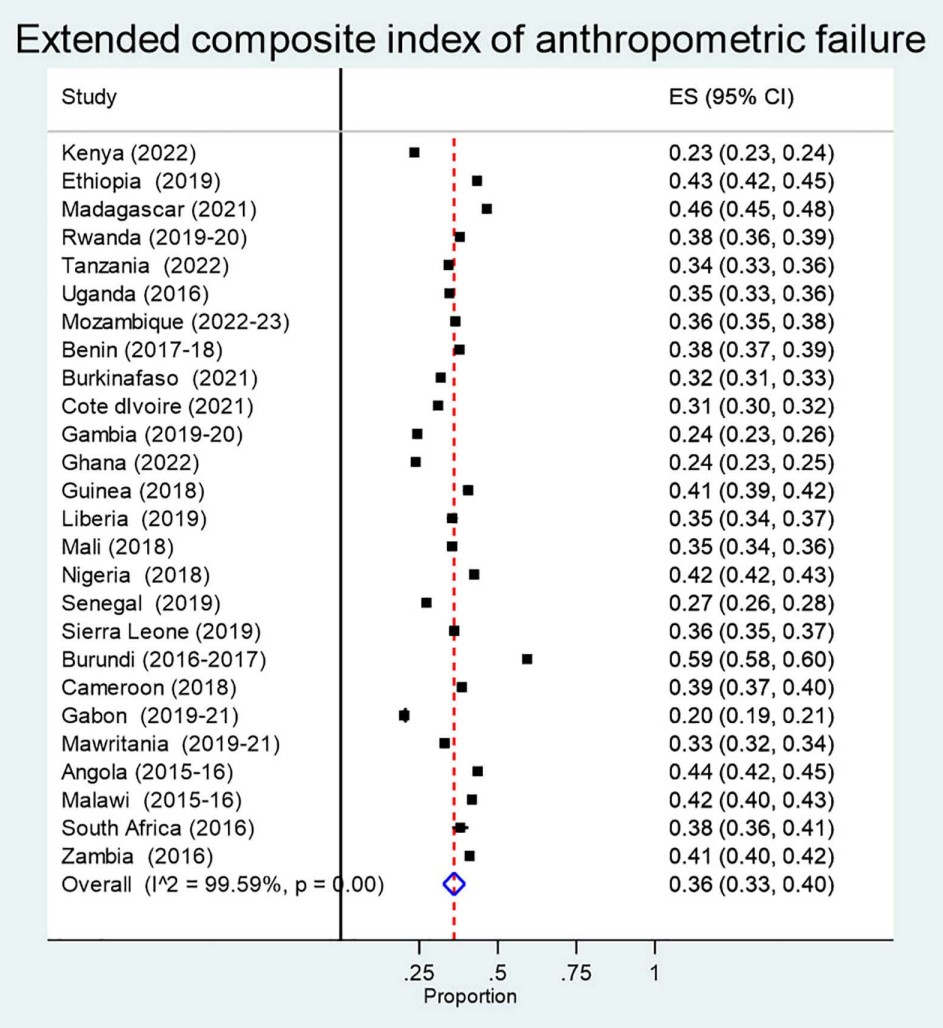

**Fig 1. Pooled prevalence of extended composite index of anthropometric failure in children under-5 years old in SSA.**

The odds of having ECIAF were higher among children whose mothers had no formal education (AOR: 1.20; 95% CI: 1.12, 1.28) and primary education (AOR: 1.36; 95% CI: 1.27, 1.45) compared with children whose mothers had secondary education or higher.

Other maternal characteristics were also significantly associated with ECIAF: children born in health institutions had lower odds of ECIAF (AOR: 0.79; 95% CI: 0.74, 0.83) compared with home deliveries; children whose mothers had antenatal care visit had lower odds (AOR: 0.83; 95% CI: 0.77, 0.90) compared with those whose mothers did not attend antenatal care; and children born to grand multiparous mothers had lower odds (AOR: 0.79; 95% CI: 0.65, 0.96) compared with those born to primiparous mothers **were however significantly protective against the odds of having ECIAF in children under-5 years in SSA.**

**Children under-5 from rich households (AOR: 1.28; 95% CI: 1.17, 1.41), middle households (AOR: 1.31; 95% CI: 1.19, 1.44), poor households (AOR: 1.42; 95% CI: 1.28, 1.56) and poorest households (AOR: 1.45; 95% CI: 1.31, 1.61), respectively had higher odds of ECIAF as compared to those from richest households.**

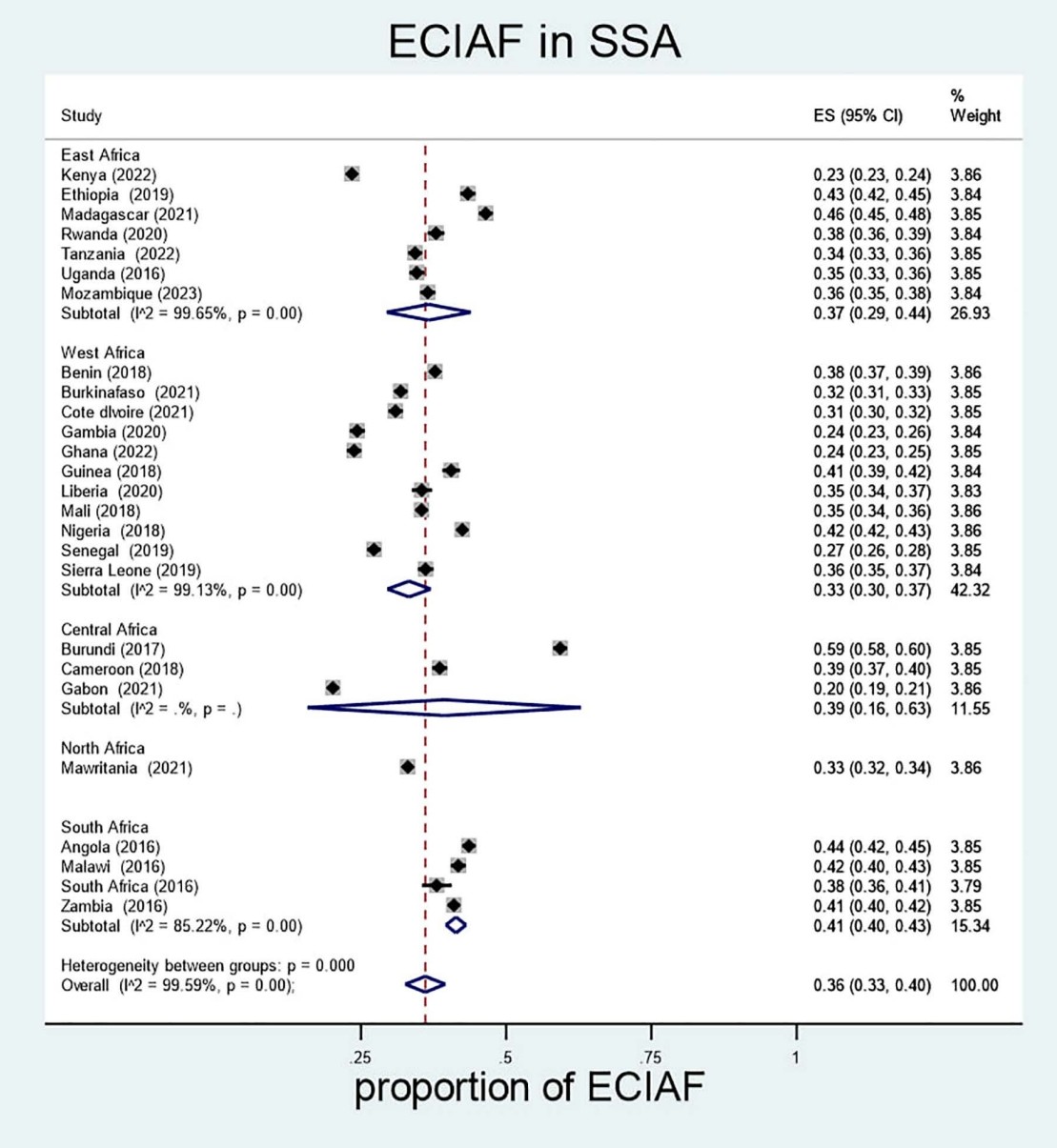

**Fig 2. Subgroups of extended composite index of anthropometric failure in children under-5 years old by African region in SSA.**

Concerning household level factors such as having 2 (AOR: 1.17; 95% CI: 1.10, 1.23) or 3–5 (AOR: 1.14; 95% CI: 1.06, 1.22) children under-5 in the household as compared to 1 under-5 child was associated with odds of having ECIAF; while having 5–9 member (AOR: 0.92; 95% CI: 0.87, 0.98) or greater than 9 family members (AOR: 0.90; 95% CI: 0.83, 0.98) as compared to 1–4 family members, was protective against the odds of having ECIAF.

With regards to the community level factors, children under-5 from Southern African countries (AOR: 2.01; 95% CI: 1.80, 2.24), Western African countries (AOR: 1.37; 95% CI: 1.24, 1.51), Eastern African countries (AOR: 1.44; 95% CI: 1.30, 1.60), and Central African countries (AOR: 1.97; 95% CI: 1.75, 2.22), respectively had significantly higher odds of ECIAF than the Northern African countries. Other factors such as living in rural residence (AOR: 1.15; 95% CI: 1.09, 1.22)

versus urban residence, and exposure to media compared no exposure to media (AOR: 1.15; 95% CI: 1.10, 1.20) exposure were significantly associated with higher odds of ECIAF in children under-5 years old in SSA (Table 4).

**Random effect analysis and model fit statistics**

The intra class correlation coefficient (ICC) estimate indicates evidence of substantial clustering which is the proportion of the total variation attributable to variation between clusters, or as correlation between children under-5 with in the same cluster (EA). The ICC in the model I (null model) indicates that about 4.00% (95% CI: 3.25%, 5.00%) of the variability in Extended composite index of anthropometric failure (ECIAF) was attributed to the community/cluster/enumeration area level variability. The MOR in the model I also revealed that if two children under-5 are taken from two different enumeration areas (EAs) i.e., one from a cluster with higher ECIAF and one from a cluster with lower ECIAF) the odds of having ECIAF among who came from cluster with higher ECIAF was 1.42 times higher as compared to their counter parts. Moreover, the PCV in the final model (model IV) showed that about 63.58% of the variability in ECIAF was explained by both community level and individual level factors. Regarding model fitness, model IV (full model) was the best fitted model since it had the lowest deviance and this model was used to assess determinants of ECIAF in children under-5 years old. All of the random effect analysis parameters favor the final model (model IV) as the best fit model (Table 5).

## Discussion

The present study primarily provides a comprehensive analysis of the prevalence and regional distribution of the Extended Composite Index of Anthropometric Failure (ECIAF) among children under-5 years old in Sub-Saharan Africa (SSA) during the Sustainable Development Goals (SDG) era. The findings revealed a pooled ECIAF prevalence of 36% (95% CI: 33%, 40%), indicating that over one-third of children in this age group experience some form of anthropometric failure.

There is limited prior research using ECIAF in the same context; however, our findings align with CIAF-based studies conducted in Ethiopia reported the prevalence of CIAF was 40.69% (95% CI: 39.41%, 42.00%) [12]. A study in Bangladesh reported a CIAF prevalence of 48.3% (95% CI: 47.1%, 49.5%) [3]. These consistent findings underscore the persistent challenge of child malnutrition in the region.

The regional disparities observed in ECIAF prevalence are noteworthy. East Africa exhibited a prevalence of 37% (95% CI: 29%, 44%), with Madagascar and Kenya showing the highest (46%) and lowest (23%) rates, respectively. West Africa's prevalence stood at 33% (95% CI: 30%, 37%), with Nigeria at 42% and both Ghana and Gambia at 24%. Southern Africa reported a prevalence of 41% (95% CI: 40%, 43%), with Angola (44%) reporting the highest, and South Africa (38%) the lowest within the region. Central Africa had a prevalence of 39% (95% CI: 16%, 63%), with Burundi at 59% and Gabon at 20%. These variations may be attributed to differences in socioeconomic status, healthcare access, and nutritional practices across regions [13].

The multilevel analysis identified a constellation of determinants operating at the child (age, birth type and size, comorbidity), maternal (education, antenatal care and place of delivery, parity), household (wealth, number of under-5 children, family size) and community levels (region, rural residence, media exposure) that together shape ECIAF risk. We now discuss the most salient patterns within each domain. Male children were more affected than females, consistent with findings from Ethiopia [12]. This may be attributed to biological and physiological differences that increase vulnerability among males in early childhood. Studies suggest that male infants have higher basal metabolic rates and energy requirements, making them more susceptible to growth faltering under adverse nutritional and environmental conditions [14]. Additionally, male children tend to have weaker immune responses during early life, increasing their risk of infections that can exacerbate undernutrition [6,15,16].

In the present study; the risk of having ECIAF was positively associated with child characteristics: child age group 6–23 months, child age group 24–59 months, multiple birth type, child size at birth average, child size at birth small, having comorbidities and negatively associated with female sex, birth interval 24 months or more, birth order 2nd or 3rd; maternal

**Table 4. Results of a multilevel logistic regression analysis shows factors associated with extended composite index of anthropometric failure among children under-5 in SSA.**

| Variable | Category | Model I | Model II | Model III | Model IV |
|---|---|---|---|---|---|
| **Child characteristics** | | | | | |
| Sex of child | Male | | 1.00 | | 1.00 |
| | Female | | 0.70 (0.67, 0.72) | | **0.69 (0.66, 0.72) *** |
| Age of child | < 6 months | | 1.00 | | 1.00 |
| | 6-23 months | | 1.53 (1.44, 1.63) | | **1.56 (1.46, 1.66) *** |
| | 24-59 months | | 1.98 (1.83, 2.13) | | **2.03 (1.88, 2.19)*** |
| Type of birth | Single | | 1.00 | | 1.00 |
| | Multiple | | 2.36 (2.03, 2.74) | | **2.38 (2.05, 2.76)*** |
| Birth interval | < 24 months | | 1.00 | | 1.00 |
| | ≥ 24 months | | 0.87 (0.82, 0.92) | | **0.86 (0.81, 0.91)*** |
| Birth order | 1st child | | 1.00 | | 1.00 |
| | 2nd or 3rd child | | 0.91 (0.86, 1.16) | | 0.90 (0.78, 1.13)* |
| | 4th or above | | 0.87 (0.78, 0.96) | | 0.87 (0.78, 0.96) |
| Child size at birth | Large | | 1.00 | | 1.00 |
| | Average | | 1.20 (1.14, 1.25) | | **1.20 (1.14, 1.26)*** |
| | Small | | 1.69 (1.58, 1.80) | | **1.80 (1.68, 1.93)*** |
| Has comorbidity | No | | 1.00 | | 1.00 |
| | Yes | | 1.15 (1.10, 1.20) | | **1.12 (1.07, 1.16)*** |
| Child vaccination status | Complete | | 1.00 | | 1.00 |
| | Incomplete | | 0.95 (0.91, 1.00) | | 0.99 (0.94, 1.04) |
| **Maternal characteristics** | | | | | |
| Education level of the mother | Secondary education and above | | 1.00 | | 1.00 |
| | Primary education | | 1.23 (1.15, 1.31) | | **1.20 (1.12, 1.28)*** |
| | No formal education | | 1.30 (1.22, 1.39) | | **1.36 (1.27, 1.45)*** |
| Age of the mother in years | 15-24 years | | 1.00 | | 1.00 |
| | 25-34 years | | 1.23 (1.15, 1.31) | | 1.02 (0.96, 1.09) |
| | 35-49 years | | 1.30 (1.22, 1.39) | | 1.06 (0.98, 1.15) |
| | Married | | 1.00 | | 1.00 |
| | Single | | 1.17 (1.03, 1.32) | | 1.07 (0.94, 1.21) |
| | Divorced | | 1.09 (0.92, 1.29) | | 1.07 (0.90, 1.28) |
| | Widowed | | 1.12 (0.93, 1.35) | | 1.08 (0.90, 1.30) |
| Place of delivery | Home | | 1.00 | | 1.00 |
| | Health institution | | 0.79 (0.75, 0.83) | | **0.79 (0.74, 0.83)*** |
| Antenatal care visit | Has no ANC visit | | 1.00 | | 1.00 |
| | Has ANC visit | | 0.82 (0.76, 0.90) | | **0.83 (0.77, 0.90)*** |
| Parity | Primi parity | | 1.00 | | 1.00 |
| | Multi parity | | 0.89 (0.76, 1.04) | | 0.90 (0.77, 1.05) |
| | Grand multi parity | | 0.79 (0.65, 0.95) | | **0.79 (0.65, 0.96)* |
| **Household characteristics** | | | | | |
| Number of children under-5 in the household | 1 child | | 1.00 | | 1.00 |
| | 2 children | | 1.18 (1.11, 1.25) | | **1.17 (1.10, 1.23)*** |
| | 3-5 children | | 1.14 (1.06, 1.22) | | **1.14 (1.06, 1.22)** |
| Number of household members | 1-4 family members | | 1.00 | | 1.00 |
| | 5-9 family members | | 0.92 (0.87, 0.98) | | **0.92 (0.87, 0.98)* |
| | ≥10 family members | | 0.84 (0.78, 0.92) | | **0.90 (0.83, 0.98)* |

*(Continued)*

**Table 4.** (Continued)

| Variable | Category | Model I | Model II | Model III | Model IV |
|---|---|---|---|---|---|
| Wealth index of the household | Richest | | 1.00 | | 1.00 |
| | Rich | | 1.36 (1.24, 1.49) | | **1.28 (1.17, 1.41)\*\*\*** |
| | Middle | | 1.47 (1.34, 1.60) | | **1.31 (1.19, 1.44)\*\*\*** |
| | Poor | | 1.66 (1.52, 1.82) | | **1.42 (1.28, 1.56)\*\*\*** |
| | Poorest | | 1.76 (1.60, 1.93) | | **1.45 (1.31, 1.61)\*\*\*** |
| **Community level characteristics** | | | | | |
| Place of residence | Urban | | | 1.00 | 1.00 |
| | Rural | | | 1.62 (1.56, 1.68) | **1.15 (1.09, 1.22)\*\*\*** |
| Media exposure | Yes | | | 1.00 | 1.00 |
| | No | | | 1.44 (1.39, 1.48) | **1.15 (1.10, 1.20)\*\*\*** |
| African region | Northern | | | 1.00 | 1.00 |
| | Southern | | | 1.40 (1.31, 1.51) | **2.01 (1.80, 2.24)\*\*\*** |
| | Western | | | 1.11 (1.04, 1.18) | **1.37 (1.24, 1.51)\*\*\*** |
| | Eastern | | | 0.98 (0.91, 1.05) | **1.44 (1.30, 1.60)\*\*\*** |
| | Central | | | 1.49 (1.37, 1.62) | **1.97 (1.75, 2.22)\*\*\*** |

NB: **Model I** is the null model; **model II** is the model with level 1 predictors; **Model III** is with level 2 predictors and **Model IV** is both level 1 & 2 predictor variables (Full model). **1** is reference category and **\*** is p-value less than 0.05(level of significance), **\*\*** is p-value less than 0.01(level of significance), **\*\*\*** is p-value less than 0.001(level of significance).

**Table 5. Random effect analysis and model fit statistics for factors associated with extended composite index of anthropometric failure among children under-5 in Sub-Saharan Africa.**

| Parameter | Model I | Model II | Model III | Model IV |
|---|---|---|---|---|
| Community level variance | 0.1362 (95% CI: 0.1103, 0.1681) | 0.0562 (95% CI: 0.0442, 0.0714) | 0.0612 (95% CI: 0.0496, 0.0754) | 0.0496 (95% CI: 0.0385, 0.0639) |
| ICC | 0.0397 (95% CI: 0.0325, 0.0486) | 0.0168 (95%CI: 0.0133, 0.0213) | 0.0183 (95%CI: 0.0149, 0.0225) | 0.0149 (95%CI: 0.0116, 0.0191) |
| MOR | 1.4199 | 1.2526 | 1.2649 | 1.2356 |
| PCV | Reference | 0.5874 | 0.5507 | 0.63583 |
| Log likelihood | −100230.82 | −36115.145 | −94992.02 | −35865.27 |
| Deviance | 200,461.64 | 72,230.29 | 189,984.04 | 71,730.54 |
| AIC | 200465.6 | 72292.29 | 190000 | 71804.54 |
| BIC | 200485.6 | 72570.8 | 190079.5 | 72136.95 |

characteristics: positively associated with having the risk of ECIAF: have no formal education, primary education *and were negatively associated with ECIAF: institutional delivery, ANC visits, and grand multiparity*; household characteristics: positively associated with the risk of having ECIAF were wealth index categories (poorest household, poor household, middle household, rich household), 2 under-5 children in the house hold, 3–5 under-5 children in the house hold and negatively associated with child ECIAF were family size 5–9 members and 10 or more members in the family; and community level characteristics: positively associated with child having ECIAF were rural place of residency, have no media exposure and African regions (Western region, Southern region, Eastern and Central regions).

Child age was a significant predictor of ECIAF, with children aged 6–23 months and 24–59 months having higher odds compared to infants younger than six months. This result aligns with findings from similar studies in SSA and other

low- and middle-income countries, where older children face greater nutritional deficits due to inadequate complementary feeding and increased susceptibility to infections [5,9]. Male children were more likely to experience ECIAF than females, consistent with research suggesting male infants have higher energy demands and are more susceptible to growth faltering [14].

Multiple births were significantly associated with higher ECIAF odds, likely due to increased caregiving demands and limited household resources. Children with short birth intervals (<24 months) also carried greater risk, emphasizing the importance of family planning interventions to improve child nutritional outcomes [17]. Furthermore, perceived birth size was a strong predictor: children perceived as smaller at birth were at significantly higher risk, corroborating evidence that low birth weight correlates with poor growth trajectories [6]. The presence of comorbidities also increased ECIAF odds, underscoring the detrimental impact of infections and illnesses on nutritional status [15,16].

Maternal education exhibited a strong inverse relationship with ECIAF. Children of mothers with no formal education or primary-level education had higher odds of ECIAF compared to those with secondary education or higher. This is consistent with findings from previous studies indicating that maternal education enhances knowledge of child nutrition and health practices [9,18]. Institutional delivery and antenatal care (ANC) attendance were protective factors against ECIAF, highlighting the role of maternal healthcare utilization in promoting child nutritional well-being [19]. Grand multiparity was associated with lower odds of ECIAF. The finding contrasts with several studies that link higher parity to increased nutritional risk due to maternal nutrient depletion or stretched caregiving capacity. One possible explanation for this divergence is that grand multiparous mothers may benefit from greater childcare experience and established social support systems, particularly in extended or multigenerational households. Further qualitative or longitudinal research is warranted to explore the context-specific dynamics influencing this association.

The household wealth index emerged as a critical determinant of ECIAF. Children from poorer households had significantly higher odds of ECIAF compared to those from the richest households. This finding is consistent with previous studies that have established the link between poverty, food insecurity, and child malnutrition [9,20]. The presence of multiple children under five also increased ECIAF risk, indicating resource competition within households [21]. Conversely, larger family sizes were protective, which may reflect collective caregiving practices in extended families [22].

Community-level factors such as residence, media exposure, and Africa geographic region significantly influenced ECIAF. Rural children exhibited higher ECIAF odds compared to urban children, consistent with research highlighting disparities in access to healthcare and nutrition services [23]. Limited media exposure also increased ECIAF odds, suggesting that health communication efforts play a crucial role in disseminating child nutrition knowledge [24]. Regionally, children in Southern, Western, Eastern, and Central Africa exhibited significantly higher ECIAF odds compared to Northern Africa, reflecting inter-regional disparities in healthcare infrastructure, dietary practices, and socioeconomic conditions [20]. These findings emphasize the need for context-specific interventions tailored to the unique challenges in each region.

### Implications of the study findings

This study underscores the multifaceted nature of child malnutrition and highlights the importance of integrated interventions addressing individual, maternal, household, and community-level factors. Strengthening maternal health services, improving family planning, and promoting socio-economic development are critical strategies for reducing ECIAF prevalence in SSA.

### Applications of the study findings

*The findings have several practical implications for child health policy and programming. It* highlights the need for targeted interventions to reduce the burden of ECIAF in Sub-Saharan Africa. These include improving maternal education, increasing access to healthcare services, and promoting family planning. Specifically, the study supports practical actions

such as: Integrating nutrition counselling into antenatal and postnatal care services; Promoting community-based health education to increase awareness of child feeding practices; Expanding conditional cash transfer programs to support poor households with children under five; Training community health workers to identify and manage anthropometric failures early.

### Limitations of the study

Cross-sectional nature of the data may limit the ability to establish causal relationship between the identified factors and the extended composite index of anthropometric failure. Cross-sectional studies provide a snapshot of data at a single point in time, making it challenging to determine the temporal sequence of events or establish a cause-and-effect relationship. Since the data source is a secondary data source, some variables like media exposure and distance to health facility were missed. Future researches should use longitudinal designs to better understand temporal relationships and causality between determinants and ECIAF. Also, study which will focus on validation of the ECIAFI is another area of research in by the future researchers. Variable like perceived size at birth, it is prone to self-report bias. Furthermore, it is essential to identify country-specific determinants to design interventions that are tailored to the nation's unique context.

### Conclusions

This study found that the prevalence of extended composite index of anthropometric failure (ECIAF) among children under-5 years in Sub-Saharan Africa remains high, with significant regional disparities. Key determinants of ECIAF included child-specific factors such as age, sex, birth type, birth spacing, perceived birth size, and comorbidity status, along with maternal factors like education level, place of delivery, and antenatal care visits. Household economic status, number of children, and community-level variables such as rural residence and media exposure also played a crucial role in determining ECIAF risk. These findings highlight the multifaceted nature of child malnutrition and the need for targeted interventions at both individual and community levels. Policymakers should prioritize strengthening maternal and child health services, particularly in rural and low-income communities, by improving access to antenatal care and health institutional deliveries. Additionally, community-based nutrition education and social protection programs should be enhanced to address socio-economic disparities and improve early childhood nutrition outcomes.

### Supporting information

**S1 Table. List of countries and survey years used for the current study titled "Unveiling the Full Picture of Malnutrition in Sub-Saharan Africa: The Extended Composite Index of Anthropometric Failure among Children Under-5 in the SDG Era".**
(DOCX)

### Acknowledgments

The authors would like to acknowledge the MEASURE DHS program for giving permission to use the data for further analysis.

### Author contributions

**Conceptualization:** Amare Muche, Aznamariam Ayres, Tilahun Dessie Alene, Robel Asaminew Mekonnen, Elsabeth Addisu, Gebeyehu Tsega, Yawkal Tsega.

**Data curation:** Amare Muche, Aznamariam Ayres, Tilahun Dessie Alene, Robel Asaminew Mekonnen, Elsabeth Addisu, Gebeyehu Tsega, Yawkal Tsega.

**Formal analysis:** Amare Muche, Aznamariam Ayres, Tilahun Dessie Alene, Robel Asaminew Mekonnen, Elsabeth Addisu, Gebeyehu Tsega, Yawkal Tsega.

**Funding acquisition:** Amare Muche, Aznamariam Ayres, Tilahun Dessie Alene, Robel Asaminew Mekonnen, Elsabeth Addisu, Gebeyehu Tsega, Yawkal Tsega.

**Investigation:** Amare Muche, Aznamariam Ayres, Tilahun Dessie Alene, Robel Asaminew Mekonnen, Elsabeth Addisu, Gebeyehu Tsega, Yawkal Tsega.

**Methodology:** Amare Muche, Aznamariam Ayres, Tilahun Dessie Alene, Robel Asaminew Mekonnen, Elsabeth Addisu, Gebeyehu Tsega, Yawkal Tsega.

**Project administration:** Amare Muche, Aznamariam Ayres, Tilahun Dessie Alene, Robel Asaminew Mekonnen, Elsabeth Addisu, Gebeyehu Tsega, Yawkal Tsega.

**Resources:** Amare Muche, Aznamariam Ayres, Tilahun Dessie Alene, Robel Asaminew Mekonnen, Elsabeth Addisu, Gebeyehu Tsega, Yawkal Tsega.

**Software:** Amare Muche, Aznamariam Ayres, Tilahun Dessie Alene, Robel Asaminew Mekonnen, Elsabeth Addisu, Gebeyehu Tsega, Yawkal Tsega.

**Supervision:** Amare Muche, Aznamariam Ayres, Tilahun Dessie Alene, Robel Asaminew Mekonnen, Elsabeth Addisu, Gebeyehu Tsega, Yawkal Tsega.

**Validation:** Amare Muche, Aznamariam Ayres, Tilahun Dessie Alene, Robel Asaminew Mekonnen, Elsabeth Addisu, Gebeyehu Tsega, Yawkal Tsega.

**Visualization:** Amare Muche, Aznamariam Ayres, Tilahun Dessie Alene, Robel Asaminew Mekonnen, Elsabeth Addisu, Gebeyehu Tsega, Yawkal Tsega.

**Writing – original draft:** Amare Muche, Aznamariam Ayres, Tilahun Dessie Alene, Robel Asaminew Mekonnen, Elsabeth Addisu, Gebeyehu Tsega, Yawkal Tsega.

**Writing – review & editing:** Amare Muche, Aznamariam Ayres, Tilahun Dessie Alene, Robel Asaminew Mekonnen, Elsabeth Addisu, Gebeyehu Tsega, Yawkal Tsega.

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
