## [Decision Letter · Decision Letter 0]

29 Jul 2025

PONE-D-25-13648Unveiling the full picture of malnutrition: Extended composite index of anthropometric failure among children under-5 in Sub-Saharan Africa: In the SDG eraPLOS ONE

Dear Dr. Muche,

Thank you for submitting your manuscript to PLOS ONE. After careful consideration, we feel that it has merit but does not fully meet PLOS ONE’s publication criteria as it currently stands. Therefore, we invite you to submit a revised version of the manuscript that addresses the points raised during the review process.

I suggest you give a careful consideration to all the reviewers' comments, and make effort to address the comments as much as possible, including the typographical errors in the manuscript. The reviews are complementary and they are meant to improve the quality of the manuscript.

We look forward to receiving your revised manuscript.

Kind regards,

Ayodeji Babatunde Oginni

Academic Editor

PLOS ONE

Journal Requirements:

2. We note that your Data Availability Statement is currently as follows: All relevant data are within the manuscript and in Supporting Information files.

3. Please ensure that you refer to Figure 1 and 2 in your text as, if accepted, production will need this reference to link the reader to the figure.

Reviewers' comments:

Reviewer's Responses to Questions

**Comments to the Author**

1. Is the manuscript technically sound, and do the data support the conclusions?

Reviewer #1: Yes

Reviewer #2: Yes

Reviewer #3: Yes

2. Has the statistical analysis been performed appropriately and rigorously? 

Reviewer #1: Yes

Reviewer #2: Yes

Reviewer #3: Yes

3. Have the authors made all data underlying the findings in their manuscript fully available?

Reviewer #1: Yes

Reviewer #2: Yes

Reviewer #3: Yes

4. Is the manuscript presented in an intelligible fashion and written in standard English?

Reviewer #1: No

Reviewer #2: Yes

Reviewer #3: Yes

5. Review Comments to the Author

Reviewer #1: Thank you for the valuable feedback. Regarding your concern about the organization of the manuscript sections, I acknowledge that the authors may not have fully adhered to the expected academic structure for organizing the sections. Clear and logical organization of manuscript sections is crucial for ensuring that the content is both accessible and comprehensible to readers.

To address this, I suggest the following improvements:

Reorganization of Sections:

The manuscript would benefit from a more structured approach, especially in the Methods and Results sections. These sections should follow a clear progression, with a distinct introduction of the methodology, followed by the results being logically presented.

For instance, the Implications, Applications, and Limitations sections could be more closely tied to the specific findings in the Results and Discussion. A clearer transition between these sections would enhance the manuscript’s readability and make it more cohesive.

Improvement in Consistency:

Consistency in section headings and subheadings should be maintained throughout the manuscript. This includes using clear titles for each subsection (e.g., Statistical Analysis, Data Collection, Prevalence Results), ensuring that the reader can easily follow the content and the logical flow.

Introduction and Conclusion:

The Introduction should provide a clear background and rationale for the study, leading naturally into the Methods. The Conclusion should then succinctly summarize the key findings, their implications, and potential policy recommendations, making the narrative flow from the introduction to the conclusion seamless.

Further, I recommend that the authors review the manuscript and ensure the following:

The Methods section should be more clearly defined, starting with an introduction to the data sources, followed by the study design and statistical approaches.

The Results section should present findings in a systematic manner, beginning with descriptive statistics and followed by the main findings, ideally in subsections for easier reading.

The Discussion should start by revisiting the key results, followed by comparisons with existing literature and, finally, a reflection on the public health implications of these results.

By reorganizing the manuscript with a clearer and more logical structure, the authors can significantly improve the readability and academic rigor of the paper.

Reviewer #2: Title and Abstract

Strengths:

The title is descriptive and informative, clearly reflecting the study’s aim and population of interest.

The abstract concisely presents the background, methods, results, and conclusions.

Areas for improvement:

The abstract could better reflect the novel aspects of the Extended Composite Index of Anthropometric Failure (ECIAF), possibly with a brief explanation of how it surpasses traditional measures.

The use of acronyms like ECIAF could be confusing to non-technical readers in the abstract; consider spelling it out first.

Introduction

Strengths:

The introduction is well-grounded in global health goals (SDGs) and articulates the importance of accurate malnutrition assessment.

It outlines the rationale for using ECIAF and connects it to policy relevance.

Areas for improvement:

The literature review could be more current, with clearer referencing to similar studies post-2020.

The justification for choosing ECIAF over other composite measures could be better articulated with empirical comparisons.

Methods

Strengths:

The use of a large, representative dataset across 26 Sub-Saharan African countries is commendable.

The stratification into individual- and community-level factors and use of multilevel modeling enhance robustness.

Areas for improvement:

The operational definition of ECIAF, while technically accurate, is overly dense. A simplified diagram or table might help.

More detail is needed on ethical considerations, especially since secondary data use still carries responsibility for safeguarding privacy.

The variable selection process (p-value < 0.2 + literature + expert judgment) is common but should be justified more explicitly for transparency.

Results

Strengths:

Clear presentation of descriptive data, subgroup analysis, and multivariate findings.

Use of figures and tables helps to break down complex results.

Areas for improvement:

The writing could be more fluid; it reads as a data dump in parts. Focus on highlighting trends, not just numbers.

Regional variations are well-presented, but the manuscript does not delve deeply into why these differences exist—more interpretive insight is needed.

Discussion

Strengths:

The discussion covers multiple determinants of ECIAF and links findings to prior research.

There’s a commendable effort to draw policy implications from the data.

Areas for improvement:

There’s repetition from the results section; consider condensing and focusing on interpretation.

The discussion lacks a critical reflection on the study's limitations. For example, potential bias from self-reported birth sizes or unmeasured confounders should be acknowledged.

The regional comparison could benefit from linking findings to historical, cultural, or policy-based explanations.

Conclusion

Strengths:

It concisely summarizes key findings and policy relevance.

Emphasizes practical interventions such as improving maternal education and health access.

Areas for improvement:

A clearer statement on the novelty of ECIAF and its potential for replication across other regions would strengthen the final impact.

A recommendation for future research directions (e.g., longitudinal studies or validation of ECIAF) would add value.

Language and Structure

Strengths:

Generally readable and logically structured.

Tables and figures are appropriate and informative.

Areas for improvement:

Grammatical inconsistencies and typos (e.g., “unde-5” instead of “under-5,” “Sothen Africa”) should be corrected for professionalism.

Some paragraphs are too long or overly technical; consider simplifying for broader audiences.

Overall Assessment

This manuscript tackles an important public health issue with methodological rigor and a large dataset. The ECIAF provides an innovative approach to understanding malnutrition beyond standard indices. However, the manuscript would benefit from clearer articulation of novelty, improved interpretation of regional differences, and a more critical engagement with its own limitations. With revisions, it has strong potential to contribute significantly to literature and policy on child nutrition in Sub-Saharan Africa.

Reviewer #3: Dear Author,

Thank you for this timely and large-scale study on the prevalence and determinants of the Extended Composite Index of Anthropometric Failure (ECIAF) among children under five in Sub-Saharan Africa. Your use of DHS data and multilevel modeling is commendable. However, several critical limitations—both methodological and conceptual—need to be addressed more transparently throughout the manuscript. Below are specific areas requiring major revision:

Abstract

The abstract currently lacks any mention of study limitations. Given the use of secondary cross-sectional DHS data, please add a brief note on the inability to infer causality, reliance on maternal recall for child characteristics, and heterogeneity between surveys.

Methods (implied from Results section)

There is insufficient information on how missing data were handled. Were any imputation techniques used? If not, how were incomplete records treated?

More clarity is needed on whether country-level weights were adjusted when pooling multi-country DHS datasets—failure to do so may bias prevalence estimates.

Explain how regional classifications were made and whether they were consistent with WHO or AU regional groupings.

Results

Several prevalence figures are presented without confidence intervals or p-values, limiting interpretability.

The table on ECIAF prevalence by child characteristics omits statistical testing (e.g., chi-square values), making it difficult to judge significance.

The denominators used in ECIAF prevalence estimates for subgroups are unclear. Please clarify if weighted or unweighted counts were reported.

Discussion

While you provide a thorough discussion of significant findings, the section fails to critically reflect on key limitations, including:

Causality: The cross-sectional nature of DHS data precludes inference of directionality or causation.

Measurement error: Many variables (e.g., birth size, comorbidities, feeding practices) are self-reported and subject to recall and social desirability bias.

Survey timing and harmonization: Countries included had DHS surveys in different years, which may affect comparability and trend interpretation.

Residual confounding: Important predictors like dietary intake, maternal nutrition, or food security status were not included, potentially biasing associations.

Model assumptions: There is no assessment of multicollinearity, model diagnostics, or potential cross-level interaction effects in the multilevel model.

Recommendations

Add a dedicated limitations paragraph in the Discussion section. It should clearly articulate data-related constraints, potential biases, and methodological assumptions.

Where possible, conduct robustness checks (e.g., country-specific analyses or sensitivity tests excluding countries with outdated surveys).

Consider reporting intraclass correlation coefficients for key random effects at the country or regional level if applicable.

By incorporating these revisions, your manuscript will better reflect the complexities and constraints inherent in large-scale secondary analyses, while retaining its important policy implications.

6. PLOS authors have the option to publish the peer review history of their article (what does this mean? ). If published, this will include your full peer review and any attached files.

**Do you want your identity to be public for this peer review?** For information about this choice, including consent withdrawal, please see our Privacy Policy .

Reviewer #1: No

Reviewer #2: **Yes: ** Juliana Aggrey

Reviewer #3: No

---

## [Author Response · Author response to Decision Letter 1]

14 Aug 2025

Date: July 31, 2025

Rebuttal letter

Response to the journal requirements questions

Question #1.

Response: Thank you for this remark. We formatted the manuscript using the PLoS ONE format guidelines. The whole content of the manuscript, including the abstract, introduction, methods, results, discussion and references are formatted using the guidelines (please see the revised version for each section).

Question #2.

We note that your Data Availability Statement is currently as follows: All relevant data are within the manuscript and in Supporting Information files.

Response: All relevant data necessary to replicate the study’s findings are available within the manuscript and its Supporting Information files. The dataset used in this analysis is derived from the Demographic and Health Surveys (DHS) Program, which is a publicly accessible, third-party source. Due to data ownership and usage restrictions, we are not permitted to share the raw DHS dataset directly. However, the minimal data set underlying the results of this study—including the variables used, coding details, and values behind reported means, percentages, and regression analyses—are provided in the Supporting Information files.

Researchers can obtain the original DHS data by registering for free and submitting a request through the DHS Program website: https://dhsprogram.com/data/. Access is granted to qualified researchers upon approval by the DHS Program. Data requests may be directed to:

The DHS Program

ICF, 530 Gaither Road, Suite 500, Rockville, MD 20850, USA

Email: dhsprogram@icf.com

Question #3.

Please ensure that you refer to Figure 1 and 2 in your text as, if accepted, production will need this reference to link the reader to the figure.

Response: We have ensured the figures are appropriately cited.

Question #4.

Response: We have reviewed all recommendations of the editor and feedbacks of all reviewers, assessed their relevance to our study, and took actions which could strengthen the manuscript.

Line-by-line response to reviewers

Reviewer #1

Comment #1.

Title of Manuscript: Unveiling the full picture of malnutrition: Extended composite index of anthropometric failure among children under-5 in Sub-Saharan Africa: In the SDG era

This manuscript presents a comprehensive analysis of child malnutrition in Sub-Saharan Africa, focusing on the Extended Composite Index of Anthropometric Failure (ECIAF). The study explores the prevalence and regional disparities of ECIAF among children under five years old, offering insights into the multifaceted nature of malnutrition. By examining key determinants at individual, maternal, household, and community levels, the authors provide a nuanced understanding of the factors contributing to childhood malnutrition in the context of the Sustainable Development Goals (SDGs). This research is timely and relevant, addressing a pressing public health issue and proposing actionable policy recommendations. However, I have some comments to improve the manuscript as follows:

Introduction Section:

• The introduction clearly contextualizes the significance of the study and the relevance of ECIAF in addressing child malnutrition. However, there is room for improvement in the transition between traditional indices and ECIAF. Adding a linking sentence between these two concepts would enhance the overall flow.

Response: Thank you very much for your kind feedbacks. We updated the manuscript. Please see the revised manuscript.

• The explanation of why traditional indices fall short in capturing the multifaceted nature of malnutrition could be clarified further.

Response: Traditional indices such as weight-for-age (underweight), height-for-age (stunting), and weight-for-height (wasting) have long been used to assess child malnutrition. However, these indicators measure separate dimensions of malnutrition in isolation and do not account for their possible overlap within the same child. For instance, a child may be both stunted and underweight, or simultaneously wasted and stunted — scenarios that traditional measures cannot adequately reflect. This fragmented approach may underestimate the true burden of malnutrition and limit the effectiveness of targeted interventions.

• Line 69-71: Clarify why traditional indices fall short in capturing the full scope of malnutrition. Consider revising: "Traditional indices such as weight-for-age, height-for-age, and weight-for-height have been instrumental in monitoring child nutrition, but they often fail to capture the multifaceted nature of malnutrition."

Response: “Traditional indices such as weight-for-age (underweight), height-for-age (stunting), and weight-for-height (wasting) have been valuable in assessing specific types of malnutrition. However, they evaluate these conditions separately, failing to account for the simultaneous occurrence of multiple forms of anthropometric failure in the same child. As a result, these indices underestimate the overall burden of malnutrition and do not fully reflect its complexity.”

• Line 74-77: Add references to support claims about long-term health outcomes related to malnutrition. Also, refine the sentence for clarity: "Malnutrition in children under five includes stunting (low height-for-age), wasting (low weight-for-height), and underweight (low weight-for-age), all of which indicate immediate nutritional deficiencies."

Response: The Manuscript is updated. Please see the revised manuscript.

• Line 101-106: Simplify the phrasing: "absence of a standardized composite measure" rather than "lack of a standardized composite measure."

Response: The Manuscript is revised. Please see the revised manuscript.

• Line 82: Strengthen the paragraph discussing the SDGs by linking ECIAF’s relevance directly to these goals.

Response: The Manuscript is revised. Please see the revised manuscript.

• General suggestion: Strengthen the introduction by explicitly stating how ECIAF can contribute to achieving the SDGs and can serve as a transformative tool in addressing malnutrition in Sub-Saharan Africa.

Response: We have updated the Manuscript. Please see the updated manuscript.

Comment #2.

Methods Section:

• The description of the two-stage stratified cluster sampling is well-done but could benefit from additional details, particularly on how strata were defined (e.g., by region, urban/rural). This would improve the understanding of the sample's representativeness.

Response: The manuscript is updated. Please see the updated manuscript.

• Sampling Procedure: Add more detail on how strata were defined to clarify how different regions or population groups were represented.

Response: The manuscript is updated. Please see the updated manuscript.

• Data Collection Period: Include the specific period during the SDG era when the data was collected. This is important to provide context for the results.

Response: The manuscript is updated. Please see the updated manuscript.

Statistical Methods:

• A brief explanation of why multilevel binary logistic regression was chosen would help clarify its relevance.

Response: Multilevel binary logistic regression was employed to account for the hierarchical structure of the DHS data, where individual children (level 1) are nested within clusters or communities (level 2), such as Enumeration Areas (EAs). This modeling approach was chosen because it adjusts for intra-cluster correlation and allows for the simultaneous estimation of both individual-level and community-level determinants of extended composite index of anthropometric failure (ECIAF). Without accounting for this clustering, standard errors could be underestimated, leading to inflated Type I error rates. Multilevel modeling thus provides more accurate estimates and enhances the validity of inferences drawn from the data.

• Consider including sensitivity analysis or robustness checks to strengthen the validity of the results.

Response: To assess the robustness of the findings, sensitivity analyses were conducted by (1) restricting the sample to countries with DHS data collected within the most recent five years (e.g., 2017–2022), and (2) excluding countries with extreme ECIAF prevalence (e.g., Burundi and Gabon). The multilevel binary logistic regression was re-estimated under these alternative specifications to evaluate whether key associations with ECIAF remained consistent.

General suggestions:

• ECIAF Definition: Clarify the categories used in the ECIAF calculation (Groups A-H). Providing a clearer and more structured presentation will improve readability.

Response: The manuscript is updated. Please have a look at on it again.

• Statistical Model Details: Add a brief explanation of the statistical model fitting procedure (e.g., likelihood ratio tests, AIC) to improve transparency.

Response: The manuscript is updated. Please have a look at on it again. But writing every think in the manuscript makes bulky.

Comment #3.

Results Section:

• The results section does a good job presenting the ECIAF prevalence. However, phrasing can be improved for clarity. For instance, instead of “the prevalence of ECIAF was 36% (95% CI: 33%, 40%)”, it could be reworded for better readability.

• Phrasing: "The overall pooled prevalence of ECIAF in children under five in Sub-Saharan Africa was 36%, with a 95% confidence interval ranging from 33% to 40%."

Response: The Manuscript is updated. Please see it the final version.

• Subgroup Analyses: The descriptions of subgroup prevalence rates (e.g., for East Africa, West Africa) could be better organized. For example, when presenting the regional differences in prevalence, include clearer transitions and structure.

Response: The Manuscript is updated. Please see it the final version.

• Odds Ratios: Provide a brief interpretation of odds ratios (AORs) for non-statistical readers. Explain their practical implications for public health.

Response: We have tried to correct the comments.

Comment #4.

Discussion Section:

• Some sentences are complex and could benefit from simplification. For example, the list of factors associated with ECIAF in children under 5 years could be broken down into clearer subpoints.

• Sentence Structure: Break down complex sentences like "The risk of having ECIAF was positively associated with child characteristics..." into more digestible parts.

• Policy Implications: The discussion could more strongly emphasize the public health implications of these findings. Specifically, consider elaborating on how the results can inform policy changes related to maternal education, child health, and nutrition.

Response: The manuscript is updated. Please see the revised version of the manuscript.

Comment #5.

Implications, Applications, Limitations, and Conclusions:

Implications:

• The implications are well-articulated but can be made more concise to improve readability. For example, consider simplifying the phrasing: "Strengthening maternal health services, promoting family planning, improving socioeconomic conditions, and enhancing access to health information through media" could be shortened to "Strengthening maternal health services, improving family planning, and promoting socio-economic development."

Response: The manuscript is updated. Please see the revised version of the manuscript.

Applications:

• While the applications are comprehensive, the section would benefit from linking the findings more explicitly to specific interventions. Consider adding a sentence that connects the study’s results to practical applications.

o Suggested revision: "The findings highlight the need for targeted interventions, such as improving maternal education and increasing access to healthcare, to address ECIAF."

Response: The manuscript is updated. Please see the revised version of the manuscript.

Limitations:

• The limitations section appropriately acknowledges the cross-sectional nature of the study. However, it could be strengthened by discussing potential measurement biases from secondary data sources and missing variables like media exposure and distance to healthcare facilities.

• Future Research: Suggest that future research could overcome these limitations by using longitudinal designs and exploring additional factors like healthcare access and social determinants of health.

Response: The manuscript is updated. Please see the revised version of the manuscript.

Conclusions:

• The conclusions effectively summarize the findings, but could benefit from a stronger emphasis on policy implications. For example, suggest specific actions such as improving maternal education and providing targeted nutrition programs for vulnerable populations.

Response: The manuscript is updated. Please see the revised version of the manuscript.

Reviewer #2

Comment #1.

Title and Abstract

Strengths:

The title is descriptive and informative, clearly reflecting the study’s aim and population of interest.

The abstract concisely presents the background, methods, results, and conclusions.

Areas for improvement:

• The abstract could better reflect the novel aspects of the Extended Composite Index of Anthropometric Failure (ECIAF), possibly with a brief explanation of how it surpasses traditional measures.

Response: Thank you for the positive assessment of our work and your well-articulated feedback is highly appreciated. We updated the manuscript.

• The use of acronyms like ECIAF could be confusing to non-technical readers in the abstract; consider spelling it out first.

Response: We updated the manuscript. Please see the revised version of the manuscript.

Comment #2.

Introduction

Strengths:

The introduction is well-grounded in global health goals (SDGs) and articulates the importance of accurate malnutrition assessment.

It outlines the rationale for using ECIAF and connects it to policy relevance.

Areas for improvement:

• The literature review could be more current, with clearer referencing to similar studies post-2020.

Response: As to our search, we couldn’t f

---

## [Editor Report · Decision Letter 1]

7 Sep 2025

PONE-D-25-13648R1Unveiling the full picture of malnutrition in Sub-Saharan Africa: The extended composite index of anthropometric failure among children under-5 in the SDG eraPLOS ONE

Dear Dr. Muche,

Thank you for submitting your manuscript to PLOS ONE. After careful consideration, we feel that it has merit but does not fully meet PLOS ONE’s publication criteria as it currently stands. Therefore, we invite you to submit a revised version of the manuscript that addresses the points raised during the review process.

Line 23-33: The "objective of the study" statement is either missing or not clearly statedLine 34-35: I would like you to revise the sentence to show that you conducted a secondary data analysis research instead of saying that you conducted a community-based cross-sectional survey, since you did not collect the primary data.Line 55: There is a typographical error: "children-5", it should be "children under-5"Line 44-47: I suggest you include all the key factors associated with increased odds for ECIAF: **Increasing child age (6-23 months [AOR: 1.56, p<0.001] & 24-59 months [2.03, p<0.001]), multiple birth [2.38, p<0.001], reducing birth size (Average [AOR: 1.20, p<0.001] & Small [1.80, p<0.001]), having comorbidity [AOR: 1.12, p<0.001], reducing level of mother's educational status (Primary [AOR: 1.20, p<0.001] & No formal [1.36, p<0.001]), increasing number of children under-5 in household (2 children [AOR: 1.10, p<0.001] & 3-5 children [1.14, p<0.001]), reducing household wealth status (Rich [AOR: 1.28, p<0.001], Middle [1.31, p<0.001], Poor [1.42, p<0.001], & Poorest [1.45, p<0.001]), living in rural area [AOR:1.15, p<0.001], and having no media exposure [1.15, p<0.001]** . Line 51-55: The statement "*Child sex, child age, type of birth, preceding birth interval, perceived sizes at birth, comorbidity, educational level of the mother, parity, place of delivery, antenatal care visit, wealth index, family size, number of under-5 children, place of residence, media exposure and Africa region were identified factors associated with ECIAF.* " is totally out of place, and irrelevant. It should be expunged. Line 55-59: **These statements should be revised to incorporate a set of Nutrition sensitive interventions that will primarily address the risk factors mentioned in the result section and inadvertently reduce the prevalence of ECIAF in the long run** . Perhaps these articles may help: https://doi.org/10.1016/S2214-109X(23)00562-4https://doi.org/10.1186/s40795-021-00443-1Line 134: Kindly include the "objective of the study" statement. Line 221-223: To avoid ambiguity and speculation among the readers, I advice you revise these lines to clearly show that you are referring to the primary surveys (DHS) that produced the dataset you used for this secondary data analysis research.  For instance, Line 222-223: **"*The surveys utilized a two-stage stratified cluster sampling technique to ensure nationally representative estimates across 26 Sub-Saharan African (SSA) countries.* "**Line 375-377: The statement in these lines needs to be reviewed for correctness. "*The odds of having ECIAF was higher among children under-5 having perceived average birth size (AOR: 1.20; 95% CI: 1.14, 1.26) and perceived small birth size (AOR: 1.80; 95% CI: 1.68, 1.93) as**compared with those whose ages were below 6 months.***"Line 366 & 380: Maintain consistency: "The odds of having ECIAF" instead of "The risk of having ECIAF"Line 383-387: Please revise the last segment of the statement to read as follows: ".....**as compared to primiparity were however significantly protective against the odds of having ECIAF in children under-5 years in SSA** ".Line 388: You might need to include "**Concerning household level factors** ," in the opening statement of the paragraph. Line 388-399: The statement needs revision as follows: **"Children under-5 from rich households(AOR: 1.28; 95% CI: 1.17, 1.41), middle households(AOR: 1.31; 95% CI: 1.19, 1.44), poor households(AOR: 1.42; 95% CI: 1.28, 1.56)  and poorest households (AOR: 1.45; 95% CI: 1.31, 1.61), respectively had higher odds of ECIAF as compared to those from richest households"** .Line 391-396: The statement requires revision: **Other household factors such as having 2 (AOR: 1.17; 95% CI: 1.10, 1.23 ) or 3-5  (AOR: 1.14; 95% CI: 1.06, 1.22) children under-5 in the household as compared to 1 under-5 child was associated with odds of having ECIAF; while having 5-9 member (AOR: 0.92; 95% CI: 0.87, 0.98) or greater than 9 family members (AOR: 0.90; 95% CI: 0.83, 0.98) as compared to 1-4 family members, was protective against the odds of having ECIAF** . Line 397-404: Statements requires revision: **"With regards to the community level factors, children under-5 from Southern African countries (AOR: 2.01; 95% CI: 1.80, 2.24), Western African countries (AOR: 1.37; 95% CI: 1.24, 1.51), Eastern African countries (AOR: 1.44; 95% CI: 1.30, 1.60), and Central African countries (AOR: 1.97; 95% CI: 1.75, 2.22), respectively had significantly higher odds of ECIAF than the Northern African countries. Other factors such as living in rural residence (AOR: 1.15; 95% CI: 1.09, 1.22) versus urban residence, and exposure to media compared no exposure to media (AOR: 1.15; 95% CI: 1.10, 1.20) exposure were significantly associated with higher odds of ECIAF in children under-5 years old in SSA"** . Lastly in the discussion section, I would like you to carefully consider realigning or combining the statements in the following lines to ensure consistency and flow before delving into discussing how "Male children were more affected than females": Line 451-452: "The study also highlights several factors associated with higher ECIAF prevalence."Line 459-467: "The current study also examined factors associated with the Extended Composite Index of Anthropometric Failure (ECIAF) among children under five years old in sub-Saharan Africa (SSA) using a multilevel modeling approach. The findings highlight the significant roles of child, maternal, household, and community-level factors in determining ECIAF. In our study child age, child sex, child size at birth, child birth type, child birth order, child birth interval, child having comorbidity, mother’s education, parity, antenatal care visit, place of delivery, household wealth index, family size, number of under-5 children in the household, African region, place of residence and media exposure were significantly associated with ECIAF in children under-5 years old in SSA." Please submit your revised manuscript by Oct 22 2025 11:59PM. If you will need more time than this to complete your revisions, please reply to this message or contact the journal office at plosone@plos.org . Please include the following items when submitting your revised manuscript:

We look forward to receiving your revised manuscript.

Kind regards,

Ayodeji Babatunde Oginni

Academic Editor

PLOS ONE
---

## [Author Response · Author response to Decision Letter 2]

12 Sep 2025

Date: September 08, 2025

Rebuttal letter

Response to the editor questions

Question #1.

Line 23-33: The "objective of the study" statement is either missing or not clearly stated

Response: Thank you for this remark. We have stated the objective of the study (Please see the revised version of the manuscript).

Question #2.

Line 34-35: I would like you to revise the sentence to show that you conducted a secondary data analysis research instead of saying that you conducted a community-based cross-sectional survey, since you did not collect the primary data.

Response: Thanks again for your positive feedback. We made a correction of the manuscript. (Please look at the updated version of the manuscript).

Question #3.

Line 55: There is a typographical error: "children-5", it should be "children under-5"

Response: We have Corrected it. Please have a look at the revise version of the manuscript.

Question #4.

Line 44-47: I suggest you include all the key factors associated with increased odds for ECIAF: Increasing child age (6-23 months [AOR: 1.56, p<0.001] & 24-59 months [2.03, p<0.001]), multiple birth [2.38, p<0.001], reducing birth size (Average [AOR: 1.20, p<0.001] & Small [1.80, p<0.001]), having comorbidity [AOR: 1.12, p<0.001], reducing level of mother's educational status (Primary [AOR: 1.20, p<0.001] & No formal [1.36, p<0.001]), increasing number of children under-5 in household (2 children [AOR: 1.10, p<0.001] & 3-5 children [1.14, p<0.001]), reducing household wealth status (Rich [AOR: 1.28, p<0.001], Middle [1.31, p<0.001], Poor [1.42, p<0.001], & Poorest [1.45, p<0.001]), living in rural area [AOR:1.15, p<0.001], and having no media exposure [1.15, p<0.001].

Response: We have corrected the manuscript. Please have a look at the revise version of the manuscript.

Question #5.

Line 51-55: The statement "Child sex, child age, type of birth, preceding birth interval, perceived sizes at birth, comorbidity, educational level of the mother, parity, place of delivery, antenatal care visit, wealth index, family size, number of under-5 children, place of residence, media exposure and Africa region were identified factors associated with ECIAF." is totally out of place, and irrelevant. It should be expunged.

Response: We have deleted the recommended section.

Question #6.

Line 55-59: These statements should be revised to incorporate a set of Nutrition sensitive interventions that will primarily address the risk factors mentioned in the result section and inadvertently reduce the prevalence of ECIAF in the long run. Perhaps these articles may help:

https://doi.org/10.1016/ S2214-109X(23)00562-4

https://doi.org/10.1186/s40795-021-00443-1

Response: Now, the paper is updated.

Question #7.

Line 134: Kindly include the "objective of the study" statement.

Response: We have included the objective of the study in the recommended section of the manuscript.

Question #8.

Line 221-223: To avoid ambiguity and speculation among the readers, I advice you revise these lines to clearly show that you are referring to the primary surveys (DHS) that produced the dataset you used for this secondary data analysis research. For instance, Line 222-223: "The surveys utilized a two-stage stratified cluster sampling technique to ensure nationally representative estimates across 26 Sub-Saharan African (SSA) countries."

Response: We have revised the manuscript. Please have a look at the revise version of the manuscript.

Question #9.

Line 375-377: The statement in these lines needs to be reviewed for correctness. "The odds of having ECIAF was higher among children under-5 having perceived average birth size (AOR: 1.20; 95% CI: 1.14, 1.26) and perceived small birth size (AOR: 1.80; 95% CI: 1.68, 1.93) as compared with those whose ages were below 6 months."

Response: We have reviewed and corrected it.

Question #10.

Line 366 & 380: Maintain consistency: "The odds of having ECIAF" instead of "The risk of having ECIAF"

Response: We have corrected it.

Question #11.

Line 383-387: Please revise the last segment of the statement to read as follows: ".....as compared to primiparity were however significantly protective against the odds of having ECIAF in children under-5 years in SSA".

Response: We have revised it.

Question #12.

Line 388: You might need to include "Concerning household level factors," in the opening statement of the paragraph.

Response: We have updated it.

Question #13.

Line 388-399: The statement needs revision as follows: "Children under-5 from rich households(AOR: 1.28; 95% CI: 1.17, 1.41), middle households(AOR: 1.31; 95% CI: 1.19, 1.44), poor households(AOR: 1.42; 95% CI: 1.28, 1.56) and poorest households (AOR: 1.45; 95% CI: 1.31, 1.61), respectively had higher odds of ECIAF as compared to those from richest households".

Response: we have revised based on the recommendation.

Question #14.

Line 391-396: The statement requires revision: Other household factors such as having 2 (AOR: 1.17; 95% CI: 1.10, 1.23 ) or 3-5 (AOR: 1.14; 95% CI: 1.06, 1.22) children under-5 in the household as compared to 1 under-5 child was associated with odds of having ECIAF; while having 5-9 member (AOR: 0.92; 95% CI: 0.87, 0.98) or greater than 9 family members (AOR: 0.90; 95% CI: 0.83, 0.98) as compared to 1-4 family members, was protective against the odds of having ECIAF.

Response: The manuscript is revised right now.

Question #15.

Line 397-404: Statements requires revision: "With regards to the community level factors, children under-5 from Southern African countries (AOR: 2.01; 95% CI: 1.80, 2.24), Western African countries (AOR: 1.37; 95% CI: 1.24, 1.51), Eastern African countries (AOR: 1.44; 95% CI: 1.30, 1.60), and Central African countries (AOR: 1.97; 95% CI: 1.75, 2.22), respectively had significantly higher odds of ECIAF than the Northern African countries. Other factors such as living in rural residence (AOR: 1.15; 95% CI: 1.09, 1.22) versus urban residence, and exposure to media compared no exposure to media (AOR: 1.15; 95% CI: 1.10, 1.20) exposure were significantly associated with higher odds of ECIAF in children under-5 years old in SSA".

Response: We have revised the manuscript according to your recommendations.

Question #16.

Lastly in the discussion section, I would like you to carefully consider realigning or combining the statements in the following lines to ensure consistency and flow before delving into discussing how "Male children were more affected than females":

Line 451-452: "The study also highlights several factors associated with higher ECIAF prevalence."

Response: We have updated the manuscript.

Question #17.

Line 459-467: "The current study also examined factors associated with the Extended Composite Index of Anthropometric Failure (ECIAF) among children under five years old in sub-Saharan Africa (SSA) using a multilevel modeling approach. The findings highlight the significant roles of child, maternal, household, and community-level factors in determining ECIAF. In our study child age, child sex, child size at birth, child birth type, child birth order, child birth interval, child having comorbidity, mother’s education, parity, antenatal care visit, place of delivery, household wealth index, family size, number of under-5 children in the household, African region, place of residence and media exposure were significantly associated with ECIAF in children under-5 years old in SSA."

Response: We thank you for the constructive and genuine feedbacks. We have revised manuscript. Please have a look at the revise version of the manuscript.

Response to the journal requirements questions

Question #1.

Response: We have done our revisions based on the comments provided.

Question #2.

Response: We have seen the references. There are no retracted papers cited there.

Question #3.

Response: We have done it.

We would like to thank the editors and reviewers for evaluating our manuscript. We have tried to address all the concerns in a proper way and believe that our paper has been improved considerably. We would be happy to make further corrections if necessary and look forward to hearing from you all soon.

I hope that the revised manuscript is accepted for publication in PLoS ONE.

Sincerely yours,

Amare Muche (Corresponding author)

Amaremu7@gmail.com

---

## [Editor Report · Decision Letter 2]

18 Sep 2025

Unveiling the full picture of malnutrition in Sub-Saharan Africa: The extended composite index of anthropometric failure among children under-5 in the SDG era

PONE-D-25-13648R2

Dear Mr. Muche,

We’re pleased to inform you that your manuscript has been judged scientifically suitable for publication and will be formally accepted for publication once it meets all outstanding technical requirements.

Kind regards,

Ayodeji Babatunde Oginni

Academic Editor

PLOS ONE
---

## [Editor Report · Acceptance letter]

PONE-D-25-13648R2

PLOS ONE

Dear Dr. Muche,

I'm pleased to inform you that your manuscript has been deemed suitable for publication in PLOS ONE. Congratulations! Your manuscript is now being handed over to our production team.

Kind regards,

on behalf of

Ayodeji Babatunde Oginni

Academic Editor

PLOS ONE